# A System Dynamics Model for Assessing Land-Use Transport Interaction Scenarios in Chennai, India

**Devi Priyadarisini K * and G. Umadevi**

Division of Transportation Engineering, Anna University, Chennai 600025, India
* Correspondence: priyadarisini123@gmail.com

**Abstract:** Purpose—rapid urbanization is dangerous to the development of society as it hinders development and makes urban living more difficult. This research aims to develop a system dynamics model for land use assessment at transport interactions in Chennai. Design/methodology/approach—the parameters of urbanization can be simulated using a system dynamics model. In this paper, the unchecked urbanization of Southern Chennai is considered as a case study; various parameters, such as population, land use, trip rate, Volume/Capacity (V/C), and Demand/Supply (D/S), are simulated for three different years: 2011, 2021, and 2031. Three scenarios are simulated: the Do-Min scenario (existing trend), partial scenario (some models are implemented), and the desirable scenario (all the proposed models are implemented). Findings—the simulation is performed using the Stella simulation tool. The results show that the proposed desirable model is highly effective in controlling both the population and other parameters. Originality/value—this study has been performed in the Indian city of Chennai, where such studies are rare. This study could help to analyze the traffic situation and make improvements to ease the urbanization issues. This research could also help to analyze the future traffic situation and make improvements to ease urbanization-related issues. Moreover, when the desirable scenario is followed in real-time, it could solve future problems that may arise from traffic congestion in Chennai.

**Keywords:** urbanization; system dynamics; transport demand and supply; trip generation; population

## 1. Introduction

The urban environment is riddled with space constraints related to residential, commercial, and industrial activities. The inefficient use of urban space may not be evident immediately, but would become apparent in the future. Hence, there must be careful urban planning, with analysis of different scenarios for long-term cases to prevent deterioration of land [1]. In most developing countries, especially India, sustainable urban development has often been neglected, leading to overcrowding and overuse of resources in urban areas. Efficient development requires solving urbanization issues, such as transportation, land use, and population control. [2]. Of all the resources, land use for transportation is concerning since the land and buildings use up much of the available space. Unchecked development in Chennai, the capital of the southern Indian state of Tamil Nādu, has rapidly converted agricultural and other vacant land types into unapproved structures for residential and commercial applications [3]. Since it is a coastal city, the development is not uniform in all directions due to the ocean barrier on the eastern side. Instead, there is a disproportionate increase in population in the western and southern parts of the city, and its outer limits. As a result, 21.54% of the farmlands are degraded in the northern part of coastal Tamil Nādu [4]. Such unapproved urbanization means that traffic volume often exceeds capacity, resulting in congestion and, as a consequence, causing time delays and pollution.

The primary importance of the study is that most cities in India are experiencing unplanned development because of urbanization. Higher demand for transportation results from both the massive migration of people from rural to urban regions, and natural

growth. Indian cities are experiencing worrisome growth in their human and vehicle populations, yet their road networks are not designed to handle the volume of traffic seen today [5].

Economic and transportation systems are mutually dependent on one another due to the interdependence of their supply and demand. The main component of a transportation system is the infrastructure that grants access to a certain degree of transportation supply. A number of models have been developed to assess the availability of transportation, and the enabling or restricting implications it has on mobility.

Thus, unplanned urbanization leads to the mixing of industrial and residential locations in a single region, which endangers human lives. Unplanned urbanization does not mean illegal construction. However, delays in setting specific guidelines by city corporations may lead to residential houses and commercial industries being built in the same region, especially in suburban regions outside large cities such as Chennai. Once the houses and industries are built, relocating the residents or companies from the regions is deemed impossible due to the significant capital costs involved. Demolition can be an option in case of illegal construction. However, since the buildings are often not illegal, no mitigating action can be taken once problems emerge.

Suburban areas experience significant physical and socioeconomic changes as cities continue to sprawl. The population growth rate of the suburban areas of Chennai is around 3.5%, higher than the city's overall growth rate of 2.54% [6]. Unregulated growth, low-quality housing, and inadequate infrastructure facilities are significant problems in the suburban areas of Chennai [7]. In the greater Chennai metropolitan region, settlements are growing rapidly due to urban sprawl. However, the development of these settlements is not of the same order. Some settlements are growing more than others; however, the road network does keep up with the same development, resulting in plethora of issues. Hence, it is essential to study the suburban areas and analyze their growth patterns, to project future growth levels and their impacts on transportation facilities.

There are lack of system dynamics-based studies specific to Indian cities, especially Chennai. Due to the unique nature of urban expansion in Indian cities, it is necessary to analyze these factors in the Indian context. This is also necessary to make predictive analyses for the future. Hence, this paper analyzes the various factors of urban expansion and how these factors affect the transportation sector and traffic in Indian cities. The paper also predicts the future growth levels of traffic, and studies their impact on the transportation sector.

The next section briefly describes the urbanization problems posed to developing countries, and the problems faced in the Indian city of Chennai. The following section also identifies research gaps, addresses the literature review and related studies, and defines the study's objectives. The research methodology, data collection, study region, and types of analysis used are described in Section 3, while Section 4 discusses the three scenarios in detail. Section 5 presents the research findings and model development, while Section 6 concludes the research and defines the potential scope of future studies.

## 2. Literature Review

A study on the river sediments in the Adyar and Cooum rivers was previously carried out by Saravanan et al. [8]. These two rivers are the main means of drainage in Chennai, and are both highly polluted. Analysis of the sediment quality shows that they are highly polluted with various hard metals. Upon comparison, Chennai's rivers were found to be the fifth-most polluted rivers in the world. This has mainly been attributed to the unplanned growth rate in the city and irresponsible management of the water bodies. The lack of strict enforcement has led to regular domestic and industrial water dumping directly into the rivers. There is also a delay in constructing adequate water treatment plants for the city; hence, most of the water from the sewers enters the rivers without any form of treatment. The growth dynamics of the Chennai Metropolitan Area (CMA) have been studied by Sekar and Kanchanamala, [7], and the factors influencing this growth have been analyzed. The

population was selected for the period between 1991 and 2001, and the policies and future development for the CMA have been reviewed.

Unchecked urbanization also led to floods in Chennai in December 2015. The prediction of runoff in the watershed of the Adyar river was studied by Sheena et al. [9] during the storm in 2005 and floods in 2015. ArcGIS has been used for analyzing the city's hydrology and the Adyar river's watershed region. The results show that a staggering 1.07 billion cubic meters of water flowed into the ocean during the final three months of 2015. The Chennai floods were also studied by Faiz Ahmed and Kranthi [10], and the damage assessment was assessed using vulnerability mapping. The damage incurred was assessed by mapping the watershed regions using data from Landsat-8 OLI, Sentinel-1, and CartoDEM-3. Geospatial techniques and spatial analyst tools were used, and it is seen that around 18% of the CMA area and 21% of the overall population were severely affected. The regions have been segregated into three types based on their vulnerability to flooding.

The above studies show that the various problems in Chennai and its urban regions are due to unplanned urbanization and t government failure to take necessary steps to curb this problem. The land along the road has developed into different types of buildings and establishments, which have further increased the traffic and the number of driveways. However, only a certain number of lanes can be built if urbanization is not checked due to a rapid increase in number of the buildings on either side. As a result, the roads cannot be widened, leading to more congestion and the deterioration of the road's capacity to move people and goods efficiently. This reduced efficiency of the road will eventually require an increase in the number of lanes, creating a vicious cycle of unchecked development [11].

Presently, system dynamics (SD) is developed by Forrester [12], which is an approach based on computer-aided design, policy analysis, feedback control theory, and model system feedback [13]. It applies emotional problems arising from different ecological systems characterized by mutual interaction, dependence on each other, circular causality, etc. [14]. In addition, the SD model has been widely utilized by sustainability professionals to facilitate detailed analysis of the interactive human and environmental systems [15]. The dynamic land use dataset is essential for numerous sectors, including environmental science, ecology, hydrology, farming, geology, and urban sprawl [16].

Among the previous literature, the SD model has been implemented widely for river basins, watersheds and regions [17,18]. However, there is still a need for SD models that adequately integrate the various physical, social, and economic factors that determine unchecked urbanization's current and future dynamics. However, few studies have focused on urbanization's impact on the population and economy. Therefore, to remove this research gap, an SD model was constructed to simulate the current conditions and future scenarios of urbanization development in suburban Chennai.

From the literature, very few studies are based only on land use- and transport-balancing concepts in their infant stage. This is especially true for developing countries such as India. Moreover, there is a lack of research in the area of balancing both infrastructure and development, and an inadequate level of work to establish the land's carrying capacity and the road network's balancing capacity.

From the previous literature, only Sheena et al. [9] and Ahmed and Kranthi [10] used Chennai as a case study. However, there has not been a focus on traffic growth related to urbanization. Saravanan et al. [8] studied the damage to the rivers. However, they did not discuss system dynamics. Though system dynamics analyses have been performed in other countries [15], there is limited literature focused on India. There is no consistent understanding of the selection of model limits and model sectors in the reviewed literature. Although specific trends appear, there is still no unified approach to validating system dynamics models using a behavioural replication test. Some knowledge gaps are present in the environmental and social effects of urban growth. To develop a sustainable smart city, a proper balance of land use and transportation should be built systematically, on which the planning of future traffic and transportation needs depends. Hence, this study intends to develop a system dynamics model using Stella simulation software.

### 3. Methodology

In this paper, a system dynamics model relating to transport and land use was built, as shown in Figure 1. The novelty in this work lay in the new attempt to determine the trips generated concerning Chennai.

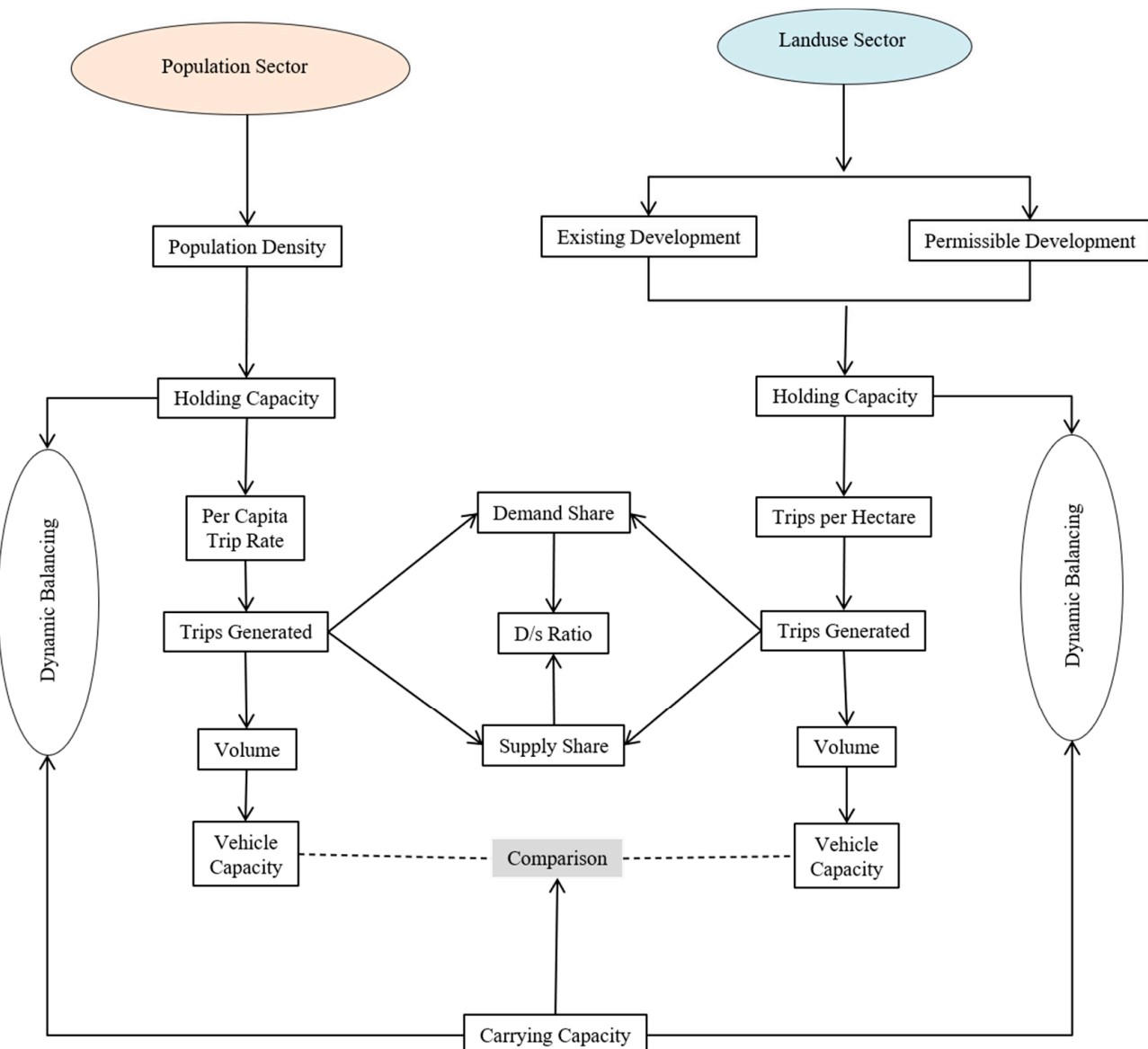

**Figure 1.** Workflow diagram.

The trips were calculated for the type of land used and compared to conventional methods. Since the geopolitical scenario and the regulation were unique in India, the study was performed from the Indian perspective to study the dynamic model's specific relevance to Chennai. Thee simulation period was taken as 20 years; the base year was 2011 and the horizon year was 2031. Data relating to vehicular growth, population, and land use in the present scenario was collected and simulated to predict future demand and supply. The holding capacity of the land use for present and future demand was simulated. The model was subjected to various scenario analyses to study the interaction between transport and land use in the study area under various conditions. The generated trips were calculated based on trip rate per hectare (TRpH) and per capita trip rate (PCTR) measures. The TRpH was the average trip rate in each hectare of the selected field, while the PCTR was the average number of trips per person. The impacts of the government policies to

achieve sustainability in transport infrastructure concerning economic development were analyzed at a macro level. A comparison of the results under various scenario options was carried out to determine the advantages of adopting the tested policy measures. Based on the results, suitable recommendations for policies were provided. The impact of these scenarios, quantified as the quality of life and level of service, were analyzed in this study. The flow of this research is shown in Figure 2.

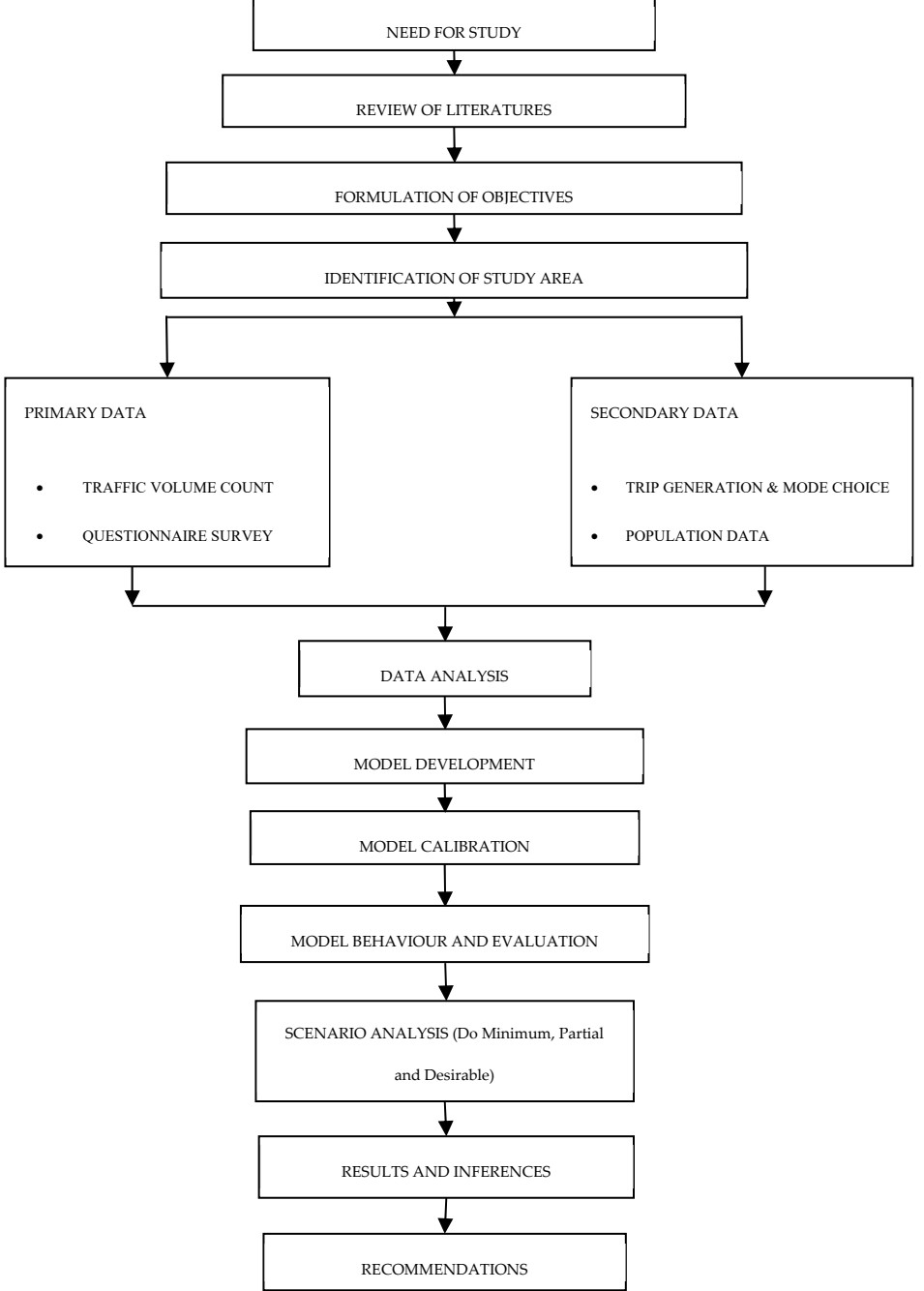

**Figure 2.** Flow of Methodology.

### 3.1. Study Area

The southern area of Chennai, surrounding Old Mahabalipuram Road (OMR) and East Coast Road (ECR), was selected as the study area. Chennai is located in the southern state of Tamil Nādu. Its location is shown in Figure 3.

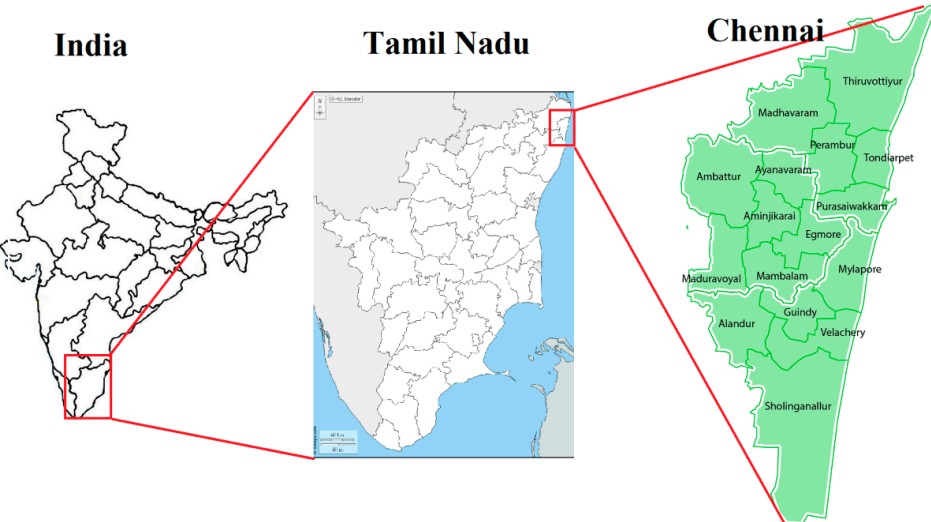

**Figure 3.** Location of the study area.

The study region is highlighted in Figure 4, and extended from Perungudi in the city's north to Kelambakkam in the city's south, as well as along the ECR and up to Nookampalayam Road to the city's west. The IT corridor is present along the OMR; hence, this region was selected by considering the adjacent regions surrounding this corridor.

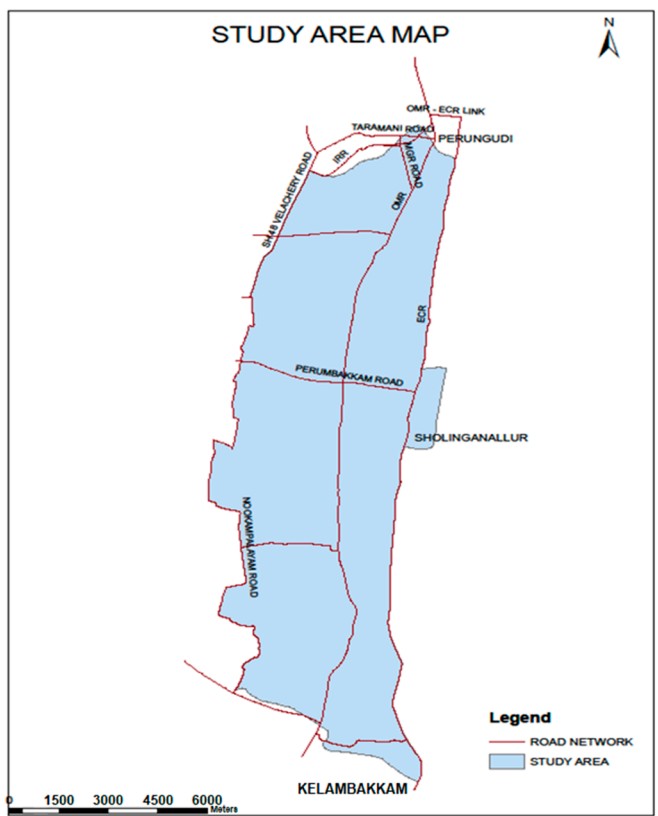

**Figure 4.** Study Area Map.

### 3.2. Data Collection and Analysis

The holding capacity of the roads was one of the main factors influencing traffic congestion. The holding capacity was determined by considering traffic volume as primary data, while demography, land use and, norms on basic infrastructures was secondary data.

Hence, both primary and secondary data had to be collected for this experiment. The collected data are briefly discussed.

### 3.2.1. Primary Data Collection

The primary data collection involved local trips bounded in the study area. It was achieved by conducting traffic volume surveys at three locations along Old Mahabalipuram Road between 3:30 p.m. and 8:30 p.m. Figure 5 illustrates the vehicle composition of the Chennai IT corridor. Vehicles were separated by type and counted for analysis. Figure 5 shows that two-wheelers and cars had the highest proportion of the total vehicle volume, at around 53% and 35%, respectively. Buses, which consisted of private, institutional, and public transportation, made up only 4% of the total vehicles. This seems to be an unhealthy proportion where environmental and level of service (LOS) considerations are concerned. Environmentally friendly vehicles, such as bicycles, made up a mere 1% of the total number of vehicles, which is an alarming statistic for a large city.

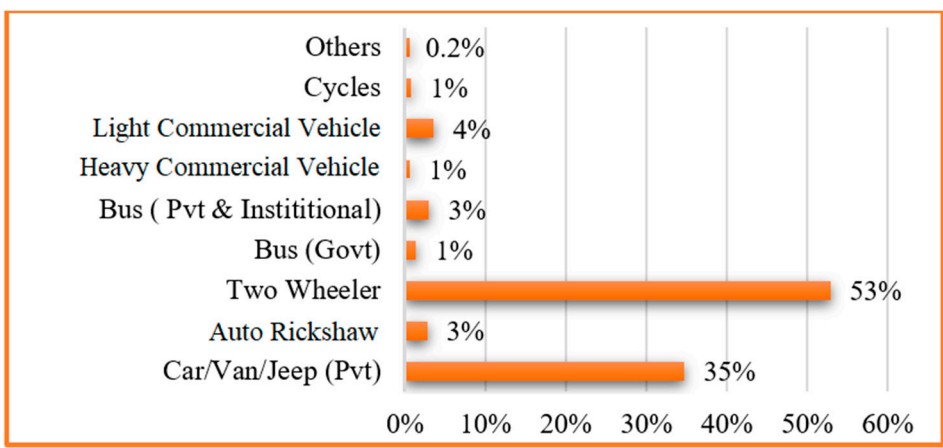

**Figure 5.** Vehicle Composition.

### 3.2.2. Trip Rate Data

The trip rate was the number of trips a person makes across a given area. The number of trips that each person takes over a certain time was also considered, and various parameters were used to measure this variable. For example, the trip rate per hectare (TRpH) was the average trip rate for every hectare in the selected region, as shown in Equation (1).

$$\text{TRpH} = \frac{\text{Total number of Trips}}{\text{Total Area in Hectares}} \tag{1}$$

Equation (1) was calculated by surveying the study area, and deriving the trips generated per hectare based on the trip data. The collected TRpH was given in Table 1.

**Table 1.** Trip rate data.

| Area | TRpH |
|------|------|
| Urban | 6000 |
| Rural | 2500 |

The term simple loop referred to a system in which one variable influences another cyclically. For example, variable A may influence variable B, which in turn influences variable C and variable D. Researchers may or may not need multiple variables to complete the loop. A positive effect was indicated using the '+' sign, while a negative effect was indicated with the '−' sign. The causal loop for all models s shown in Figure 6. It can be seen that the population variable had a positive impact on four other variables, which were in-migration, out-migration, birth and death. In contrast, the inverse had a negative impact.

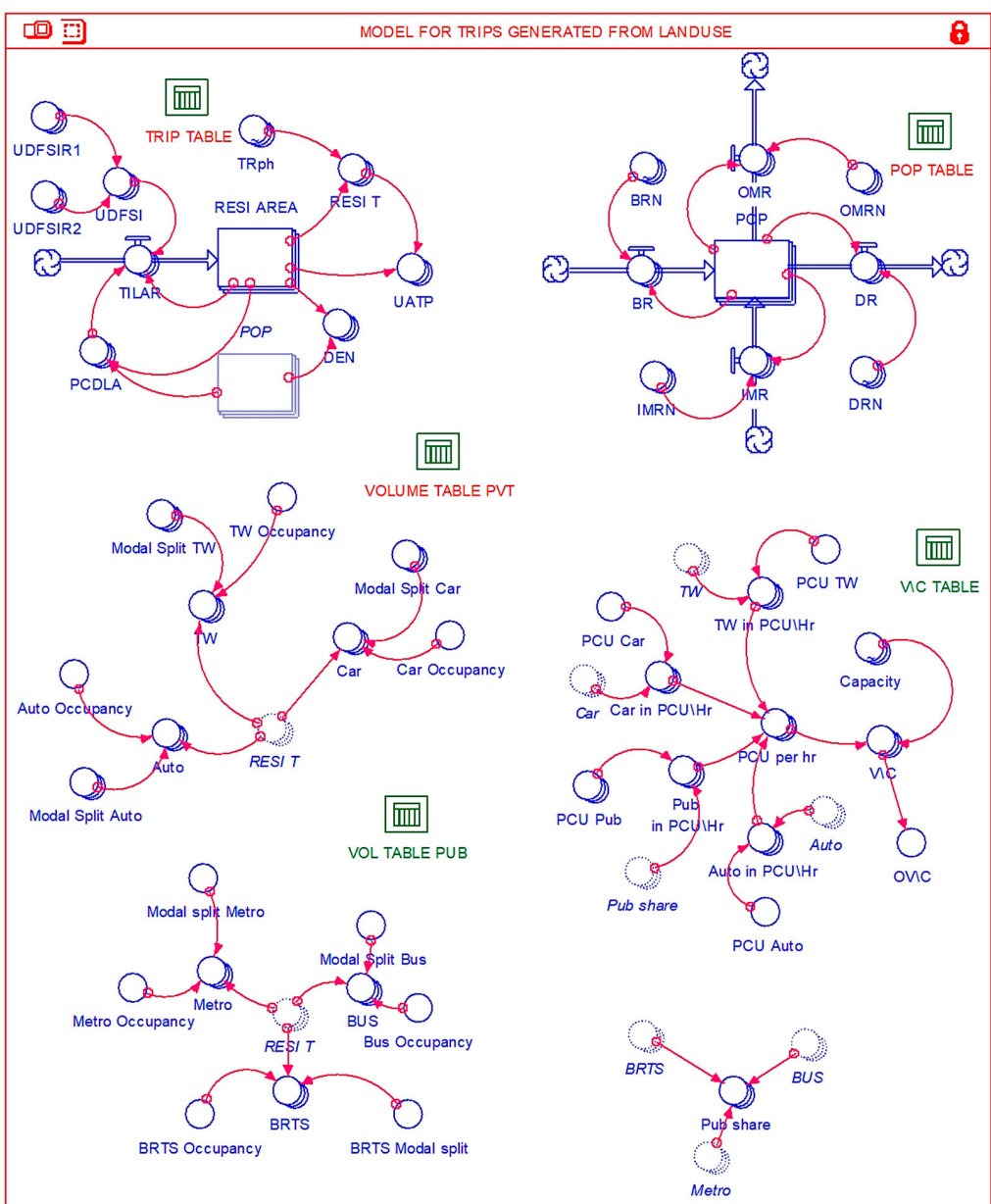

**Figure 6.** Model for trips generated.

### 3.2.3. Land Use Data

Land use was defined as the management and modification of existing land into a different type of land. In this case, it is the conversion of agricultural, forested, and wasteland into urban land. The conversion of land use must be productive in a planned urban region, given the limited land available for urban use. This paper extracted the land area corresponding to every land use type from Geo referencing. A land use map was created using ARCGIS 10.2.2, and the classifications were specified via Erdas 2014 and using LISS III data from Bhuvan [19]. The urban and rural regions were split into two sections: the areas located from Perungudi to Sholinganallur were regarded as urban areas, while the areas between Siruseri to Kelambakkam were regarded as rural regions. The land use data for both rural and urban regions from satellite data in 2021 is given in Figures 7 and 8, while the composition for the residential area is given in Table 2.

**Table 2.** Residential land area composition.

| Detail | Urban in Hec | Rural in Hec |
|---|---|---|
| Total area | 6253 | 5286 |
| Residential area | 1338 | 1721 |
| Residential area as % | 21.39% | 32.55% |

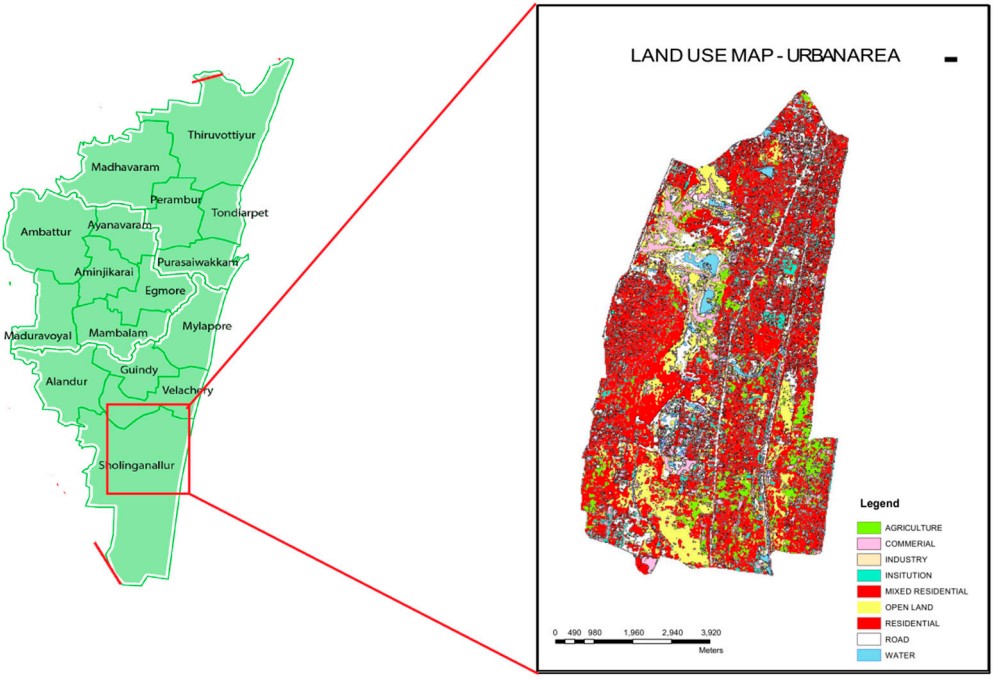

**Figure 7.** Land use map for the urban area.

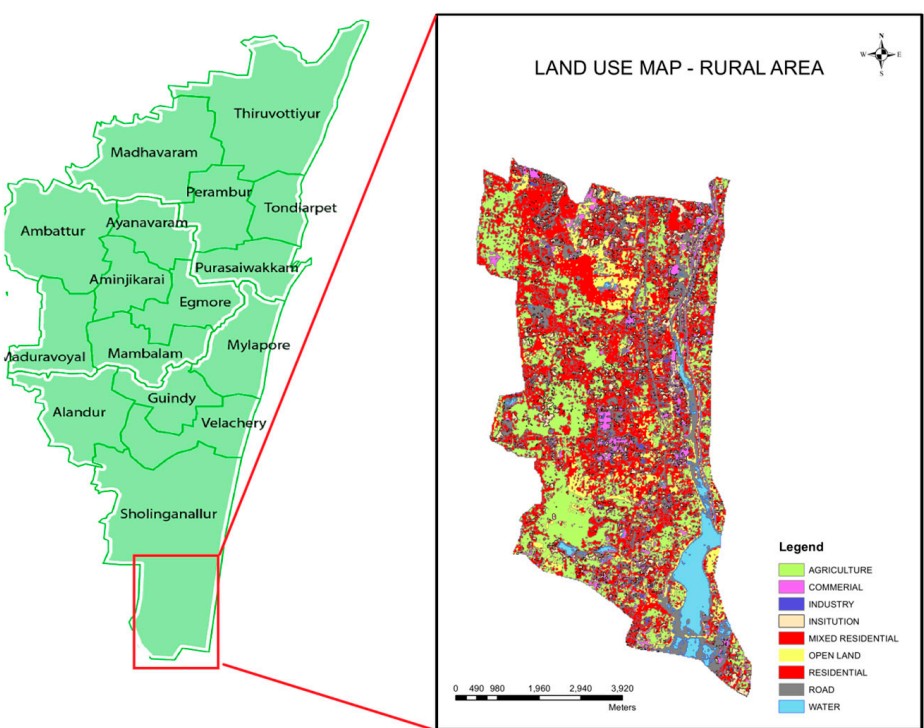

**Figure 8.** Land use map for the rural area.

Figures 7 and 8 show that the urban region consisted mainly of residential buildings, and a mixture of residential and other buildings. The total residential portion of urban land was around 21.39%. On the other hand, the rural region had more agricultural land and contained a larger area of water bodies. The percentage of residential land was comparatively higher than urban at 32.55%. The availability of open land was negligible in both regions.

### 3.2.4. Secondary Data Collection

The demography data for rural and urban areas was collected as secondary data from the 2011 Indian Census. This data defined of nature of the local population, and is quantitative in nature.

### 3.3. System Design

Stella, a tool for system dynamics, was used in this work. Stella is an object-oriented programming environment, where models can be designed by linking different parameters to create a single framework, which makes the structure of the model become clear. The model integrated various data collected through primary or secondary data, such as population, births, deaths, birth rate, and death rate. The tool also created the difference equations automatically based on these inputs.

### 4. Scenarios

Three different scenarios were considered for predicting the variables for the years 2021 and 2031. They are do minimum (Do-Min), partial, and desirable. The parameters measured were the population growth, land use, etc.

In the Do-Min scenario, the existing trend was allowed to continue with no novel innovations or implementations. With the current trend, the various parameters, such as land use and population, were measured. The next scenario was known as partial, where 50% of the new proposals were assumed to be implemented soon. The proposed desirable scenario simulated the implementation of all the proposals, and the perspective parameters were measured for the future. In the desirable method, the population density was maintained at 500 people per hectare based on the holding capacity of each region. The parameters for birth rate, death rate, in migration rate and out migration rate are shown in Table 3.

**Table 3.** Parameters for population.

|  | **Do-Min** | **Partial** | **Desire** |
|---|---|---|---|
| Birth rate per 1000 | 21–18% | 21–16% | 21–10% |
| Death rate per 1000 | 10–8% | 10–6% | 10–5% |
| In migration rate | 0.01–0.02 | 0.01–0.025 | 0.01–0.03 |
| Out migration rate | 0.001 | 0.001 | 0.001 |

The floor space index (FSI) was fully developed and fixed at 2.5. Land growth was considered only horizontally for Do-Min, while both vertical and horizontal spread was considered for the other two scenarios, as shown in Table 4.

**Table 4.** Parameters for land use.

| **Parameters** | **Do-Min** | **Partial** | **Desirable** |
|---|---|---|---|
| Land use growth rate | 0.02 | 0.02 | 0.02 |
| FSI | NA | 1–1.5 | 2.5 |

In the transport sector, an elevated corridor was proposed in the study area along the OMR to increased capacity; simultaneously, public transport usage was doubled, and

private Transport was reduced by 50%. In the desirable scenario, the incorporated elevated corridor increased demand for public transport through the introduction of metro and monorail, and reducing the private vehicle usage in a phase-wise manner. The desired split up for the three scenarios is shown in Table 5. Per capita trip rate (PCTR) for the Do-Min, partial and desirable scenarios were 1.2, 2.5 and 3, respectively. The capacity of the roads, as per the Indian Road Congress (IRC), was considered and implemented, with 5400 m for the Do-Min. An additional elevated road was added in the partial scenario, increasing capacity to 2400 m. In the desirable scenario, around 1200 m of existing roads were widened, thereby increasing the total road capacity.

**Table 5.** Split-up of vehicles.

| Parameters | Do-Min | Partial | Desire |
|:---:|:---:|:---:|:---:|
| Two-wheeler | 53 | 25 | 15 |
| CAR | 35 | 25 | 10 |
| BUS | 5 | 30 | 50 |

## 5. Model Development

### 5.1. Population Sector

The model for the population sector (shown in Figure 9) is designed to estimate the projected population. The parameters considered are birth rate, death rate, in-migration rate, and out-migration rate. The model structure was built using an array, and the results are tabulated in Table 6.

**Table 6.** Study results of population model.

| Pop | DO-MIN | | | PARTIAL | | | DESRIABLE | | |
|:---:|:---:|:---:|:---:|:---:|:---:|:---:|:---:|:---:|:---:|
| **Year** | **2011** | **2021** | **2031** | **2011** | **2021** | **2031** | **2011** | **2021** | **2031** |
| Urban limit | 249,274 | 460,993 | 834,994 | 249,274 | 442,391 | 77,650 | 249,274 | 444,484 | 785,029 |
| Rural limit | 30,951 | 25,253 | 20,353 | 30,951 | 31,678 | 32,574 | 30,951 | 119,133 | 452,593 |

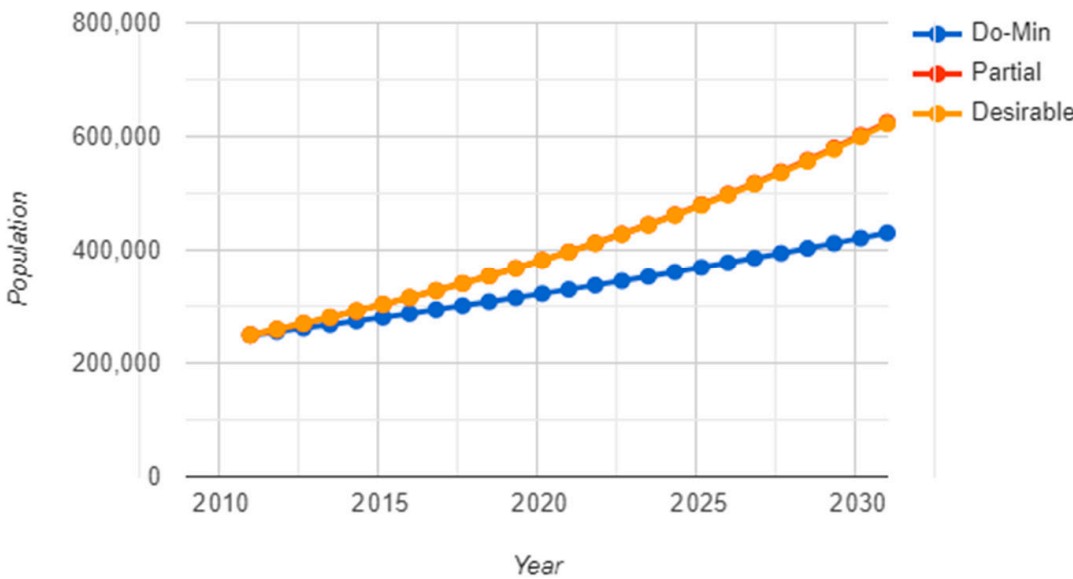

**Figure 9.** Population of the urban area for the different scenarios.

The population projection for the different scenarios is shown in Figures 9 and 10. The projection for the urban limit is shown in Figure 9. It is shown that the population in the urban regions increases rapidly in all three scenarios. The population increase is higher for the desirable scenario compared to the Do-Min. In Figure 10, while the rural population grows linearly in the first scenario, it is eventually halted and curves to a lower value in the desirable scenario. The figure for the partial scenario is almost same as for the desirable in both cases. This is a negative sign for a city since development only takes place in certain regions. Drastic reductions in population will de-stabilize the economy, leading to a crisis.

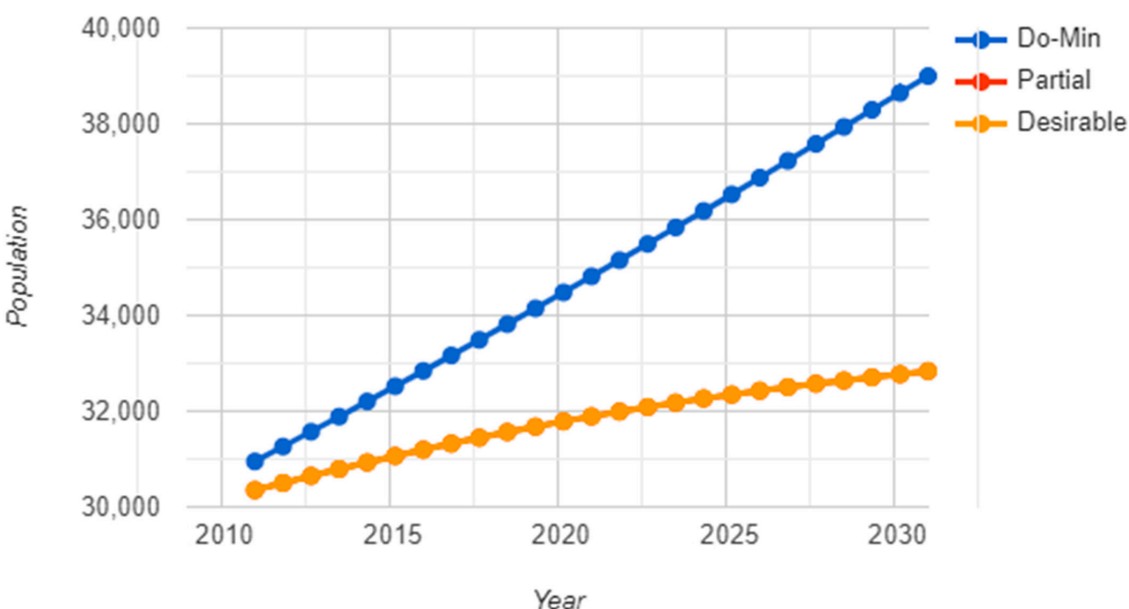

**Figure 10.** Population of the rural region for the different scenarios.

### *5.2. Trip Generation*

The frequency of trips that start and terminate in each region is modelled as trip generation. The goal of trip generation is to simulate the total number of anticipated trips to each zone within the research area. The trip generation, or trip rate, can be calculated from the land use and population data [20].

### 5.2.1. Trip Generation from Land Use

This is a novel approach to trip generation, and is built by considering existing residential land area and future developable floor area. The residential growth rate in the land area for the projected years is calculated, along with the average trip generated per hectare with the help of survey data [21]. The major variable that affects the other variables is the trip rate. The factors affected include land use, population, trip per hectare, trip rate per capita, public and private transport, IPT, and other modes of transportation. The study results are shown in Table 7. The existing residential floor area for every year can be developed with respect to the growth rate of residential intensity and the available residential area of each zone. The undeveloped floor surface index ratio (UDFSIR) is calculated using Equations (2) and (3); the former equation considers the remaining land from developed regions, while the latter equation considers remaining land from the undeveloped regions.

$$\text{UDFSIR1} = \text{Existing residential area (Remaining \% from already developed)} * \text{Plot coverage for constructing house} * \text{Undeveloped remaining FSI} \tag{2}$$

$$UDFSIR2 = \text{Existing residential area (\% yet to be developed)} * \text{Plot coverage for} \\ \text{constructing house} * \text{Undeveloped FSI} \tag{3}$$

**Table 7.** Study results for trip generation from land use.

| Trip Generation | | Do-Min | | | Partial | | | Desirable | | |
|---|---|---|---|---|---|---|---|---|---|---|
| | Year | 2011 | 2021 | 2031 | 2011 | 2021 | 2031 | 2011 | 2021 | 2031 |
| Urban | Area (hectares) | 1338 | 1447.5 | 4668.81 | 1338 | 9798 | 13174 | 1338 | 3362.74 | 7480 |
| | % increase | | 8.18% | 222.54% | | 632.29% | 34.46% | | 151.33% | 122.44% |
| | Resi trip (1000 km) | 9750 | 9790 | 10496 | 9750 | 10015 | 10311 | 9750 | 10015 | 10015 |
| | % increase | | 0.41% | 7.21% | | 2.72% | 2.96% | | 2.72% | 0% |
| Rural | Area (hectares) | 1721 | 3118.84 | 7109.51 | 1721 | 2658 | 6735 | 1721 | 2126.94 | 6215 |
| | % increase | | 81.22% | 127.95% | | 54.45% | 153.39% | | 23.59% | 192.20% |
| | Resi trip (1000 km) | 6000 | 7500 | 9000 | 6000 | 6500 | 7000 | 6000 | 6000 | 6000 |
| | % increase | | 25% | 20% | | 8.33% | 7.69% | | 0% | 0% |

The projected changes in the extent of residential areas for each scenario are shown in Figures 11 and 12. In the urban region, the residential area increased widely in the partial scenario. Residential growth is more limited in both the Do-Min and desirable scenarios, as shown in Figure 11. On the other hand, there is little difference between the three scenarios' predictions for residential growth in the rural setting, as shown in Figure 12.

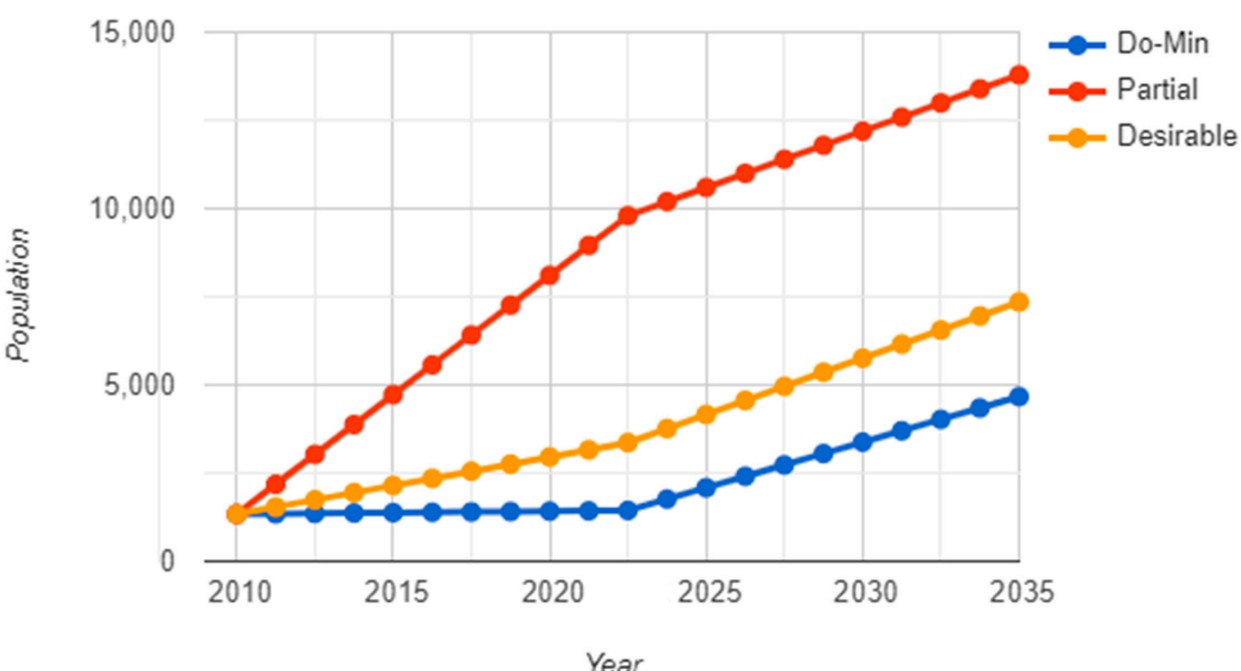

**Figure 11.** Projection of change in the extent of the residential area for the urban region.

The projected change in the trip rate per hectare for each scenario is shown in Figures 13 and 14. In the urban region, the trip rate increases significantly in the Do-Min and partial scenarios, while this increase is more controlled in the desirable scenario. After the year 2021, the trip rate starts to stabilize in the desirable scenario, as shown in Figure 13. In the rural setting, the TRpH increases from 6000 to 9000 trips for the Do-Min scenario. However, this trip rate remains constant in the desirable stage, as shown in Figure 14.

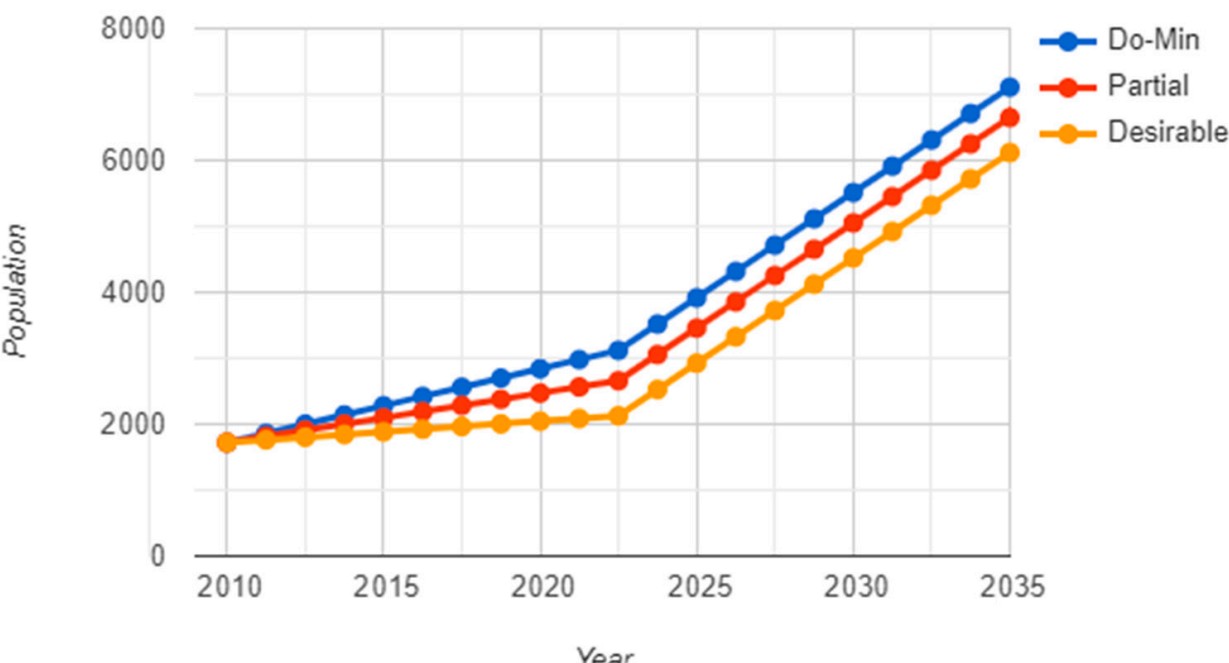

**Figure 12.** Projection of change in the extent of the residential area for the rural region.

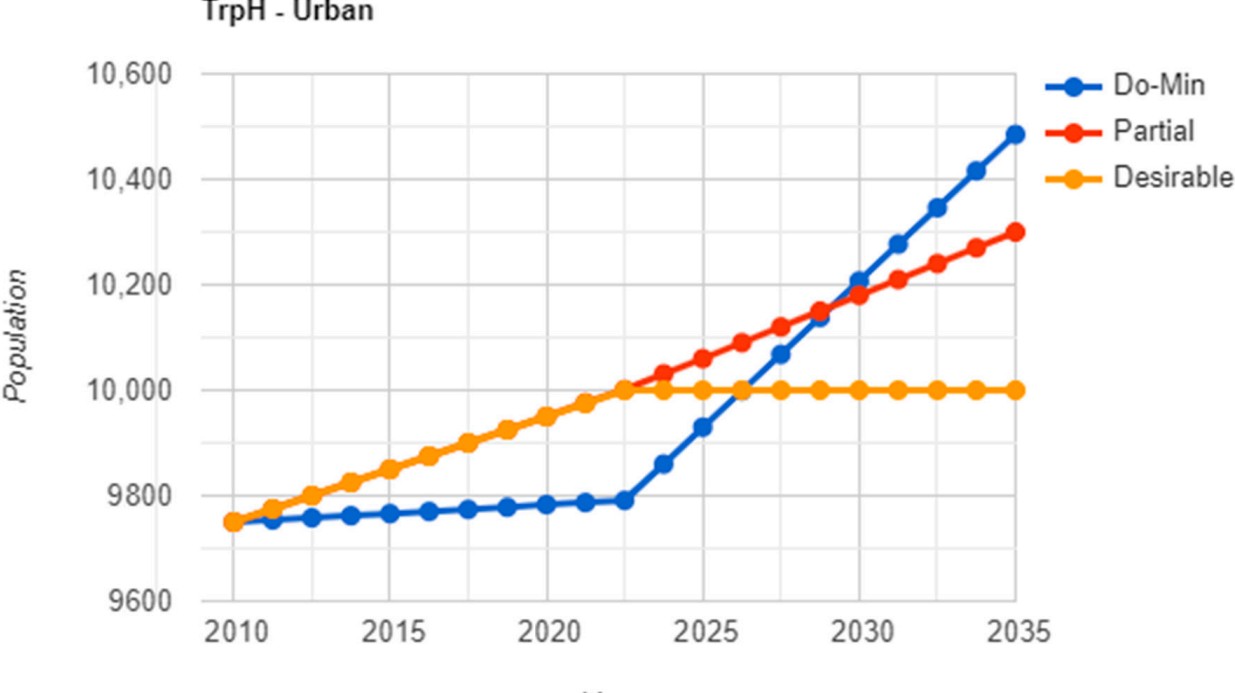

**Figure 13.** Projected change in TRpH for the urban region.

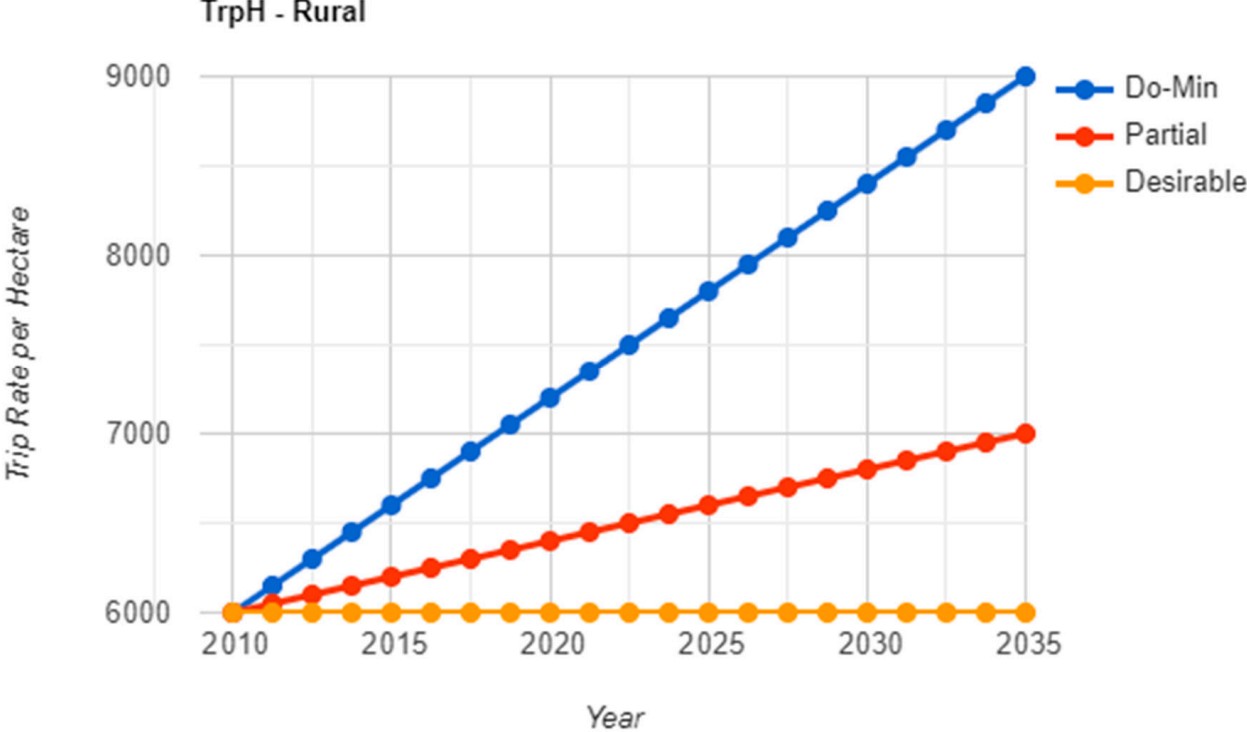

**Figure 14.** Projected change in TRpH for the rural region.

5.2.2. Trip Generation from Population

Trip generation from the population is a conventional method, and developed by per capita trip rate to population. The per capita trip rate (PCTR) is taken from the second master plan of Chennai [22]. The total vehicle trips (TVT) are calculated, which are split into total motorized trips (TMT) and total non-motorized trips (TNT). The study results for the years between 2011 and 2031 are given in Table 8.

**Table 8.** Study results for trip generation from population.

| Trip Generation | | Do-Min | | | Partial | | | Desirable | | |
|---|---|---|---|---|---|---|---|---|---|---|
| Year | | 2011 | 2021 | 2031 | 2011 | 2021 | 2031 | 2011 | 2021 | 2031 |
| Urban | TMT | 219,859 | 14,204,914 | 67,565,745 | 190,305 | 13,327,622 | 71,490,222 | 190,305 | 10,787,032 | 40,529,131 |
| | TNT | 94,225 | 6,087,820 | 28,956,747 | 126,870 | 8,885,081 | 47,660,148 | 11,160 | 70,428 | 2,996,478 |
| Rural | TMT | 12,999 | 698,117 | 2,469,064 | 11,160 | 723,936 | 3,288,042 | 126,870 | 7,191,355 | 27,019,420 |
| | TNT | 5571 | 299,193 | 1,058,170 | 7440 | 482,624 | 2,192,028 | 7440 | 466,952 | 1,997,652 |

The total motorized trips and non-motorized trips for the different scenarios are illustrated in Figures 15–18. The desirable model aims to reduce the total motorized trips rate in the urban region, while increasing it in the rural region, to have a comparatively equal increase in urban development. It can be seen from Figure 15 that there is a large variation in the trip increase between the Do-Min and the desirable scenarios. Similarly, in Figure 16, the Do-Min and partial scenarios have limited growth since most of the population h moved to the urban region; in the desirable scenario, there is reduced migration to urban regions resulting in a large increase in the motorized trip rate in the urban setting.

This is also true for non-motorized trips. As seen from Figures 17 and 18, there is a large variation in the trip increase between the Do-Min and the desirable scenario for the urban setting; in the rural setting, the Do-Min and partial scenarios have limited growth, while and desirable scenario has a larger rate f growth. This is, once again, attributed to the reduction in migration to the urban region.

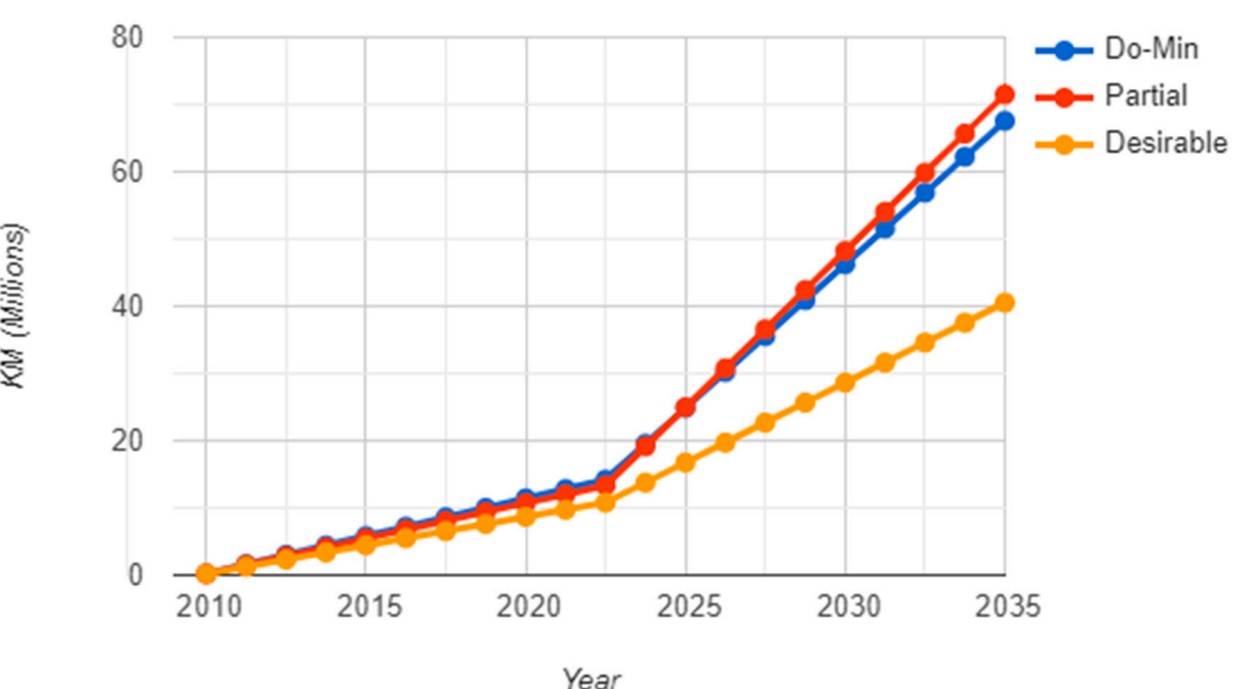

**Figure 15.** TMT for the urban region.

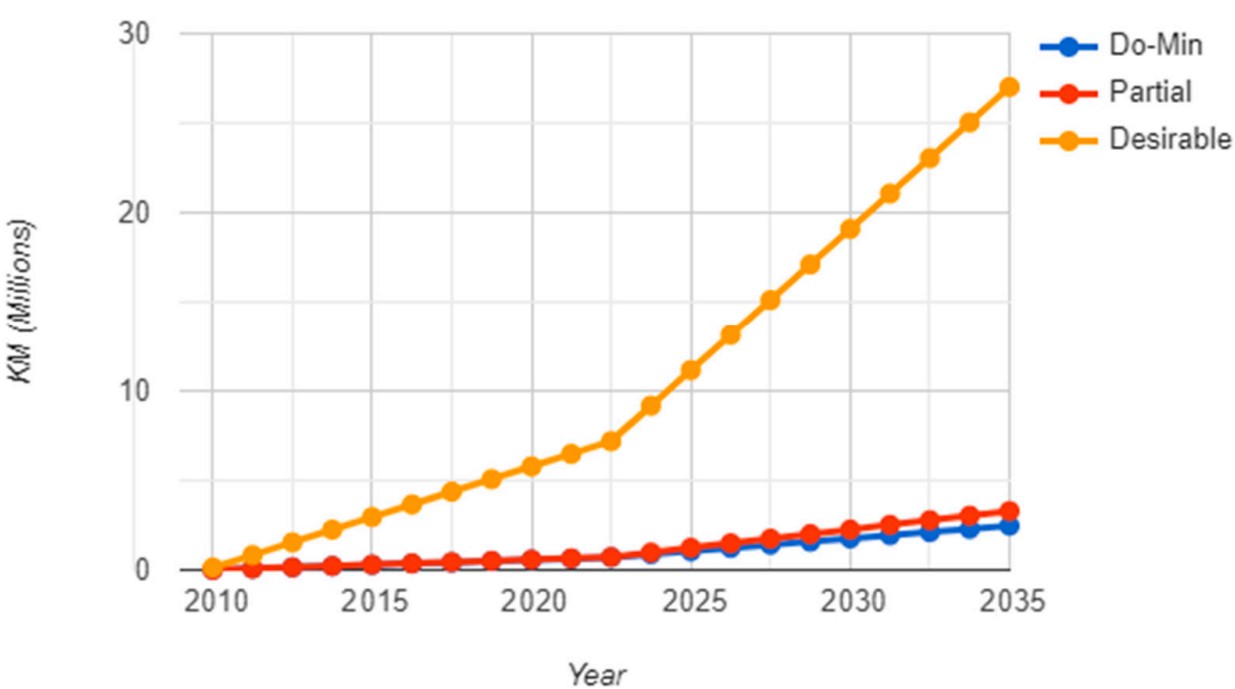

**Figure 16.** TMT for the rural region.

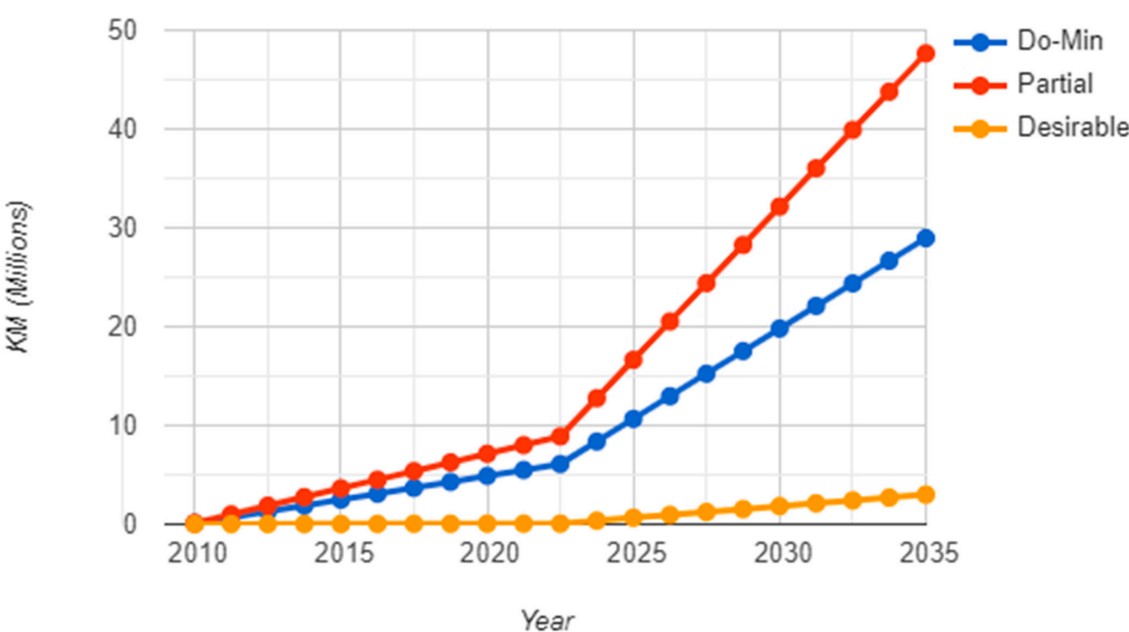

**Figure 17.** TNT for the urban region.

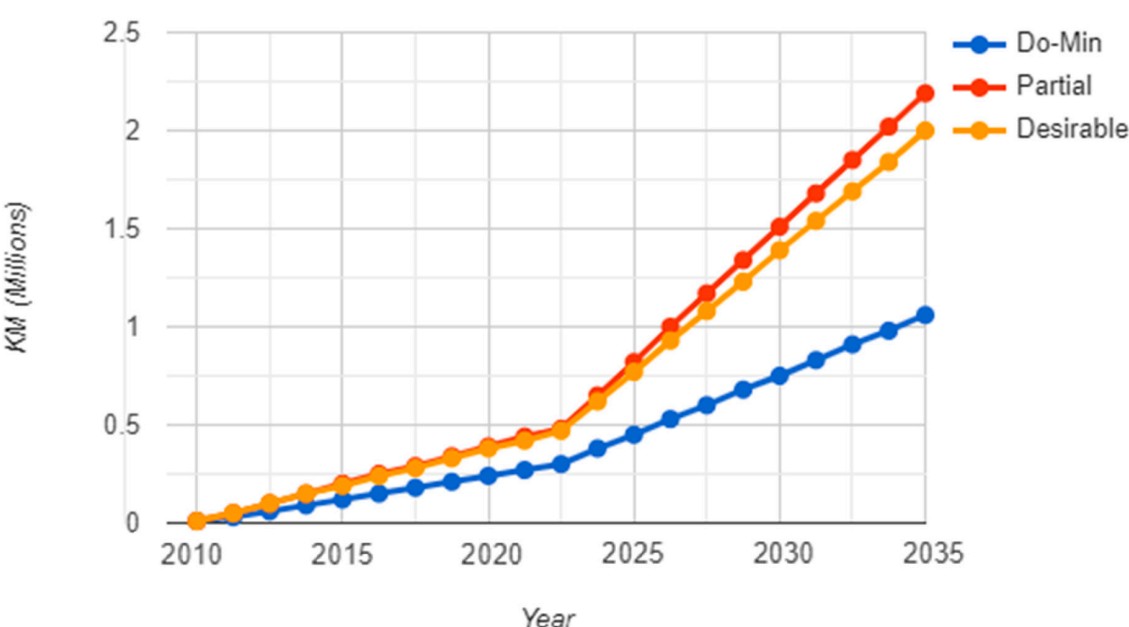

**Figure 18.** TNT for the rural region.

This finding is true for both TMT and TNT. The increase in these trips is justified since they are directly proportional to the population of the region. The population is expected to increase in both the urban and rural regions. The other factor is the increase in vehicular usage as the economic conditions of the people increase. In the Do-Min and partial scenarios, a high influx of people move into the city, thereby increasing the number of vehicles. In the desirable scenario, the rate of migration is controlled; hence, there are a more limited number of vehicles.

### 5.3. V/C Calculation

The selection of an appropriate exposure measure is a critical issue in traffic accident investigation. Currently, traffic volume is employed as a metric of visibility. The number of accidents per million vehicle kilometres is used to measure the risk of using a certain stretch of road. Assuming that traffic accidents grow in proportion to traffic flow is an oversimplified assumption that is incorrect as a rule. Using the same traffic volume on various capacity road sections provides varied operation circumstances and, hence, variable accident possibilities. A more effective measure of exposure may be the volume-to-capacity (V/C) ratio, which is one of the characteristics that affect the level of service, since there is a limit in the vehicle capacity of a particular road. V/C is a ratio that compares the actual volume of the vehicles to the maximum vehicle capacity. Ideally, This ratio should be low to ease the flow of traffic. The volume of each model of vehicle is calculated by computing the percentage modal share of each vehicle, and the occupancy ratio of the respective vehicle. The model for the V/C sector is built by accounting for the present vehicular volume and the capacity of the corridor six-lane road, as per government regulations, to determine the V/C ratio. The V/C results are shown in Table 9.

**Table 9.** Study results of V/C calculation.

| V/C | Do-Min | | | Partial | | | desirable | | |
|---|---|---|---|---|---|---|---|---|---|
| Year | 2011 | 2021 | 2031 | 2011 | 2021 | 2031 | 2011 | 2021 | 2031 |
| From LU | 0.32 | 0.67 | 1.28 | 0.32 | 1.29 | 3.14 | 0.32 | 0.56 | 0.56 |
| From Pop | 0.01 | 0.79 | 3.69 | 0.01 | 0.039 | 2.04 | 0.01 | 0.25 | 0.96 |

The ratio is predicted from two different parameters—land use and population—as shown in Figures 19 and 20. The desirable V/C ratio from the land use and population is very small, and this ratio must be maintained in the future projection. From Figure 19, the V/C projection for the first two scenarios increases to a value above one, while it remains far below one for the desirable scenario.

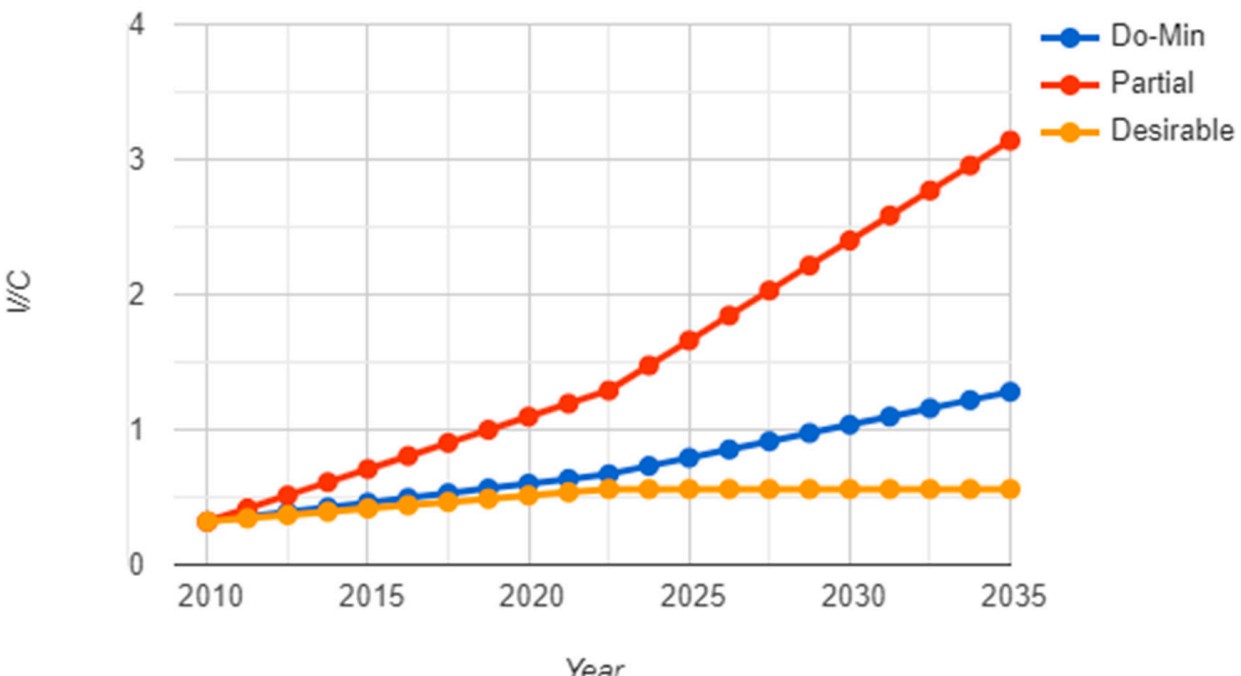

**Figure 19.** V/C ratio results for from land use.

Similarly, where the population is concerned, the desirable scenario shows favorable projections, as shown in Figure 20. The Do-Min increases from 0.1 to 3.69, while the ratio stays below 1 for desirable scenario.

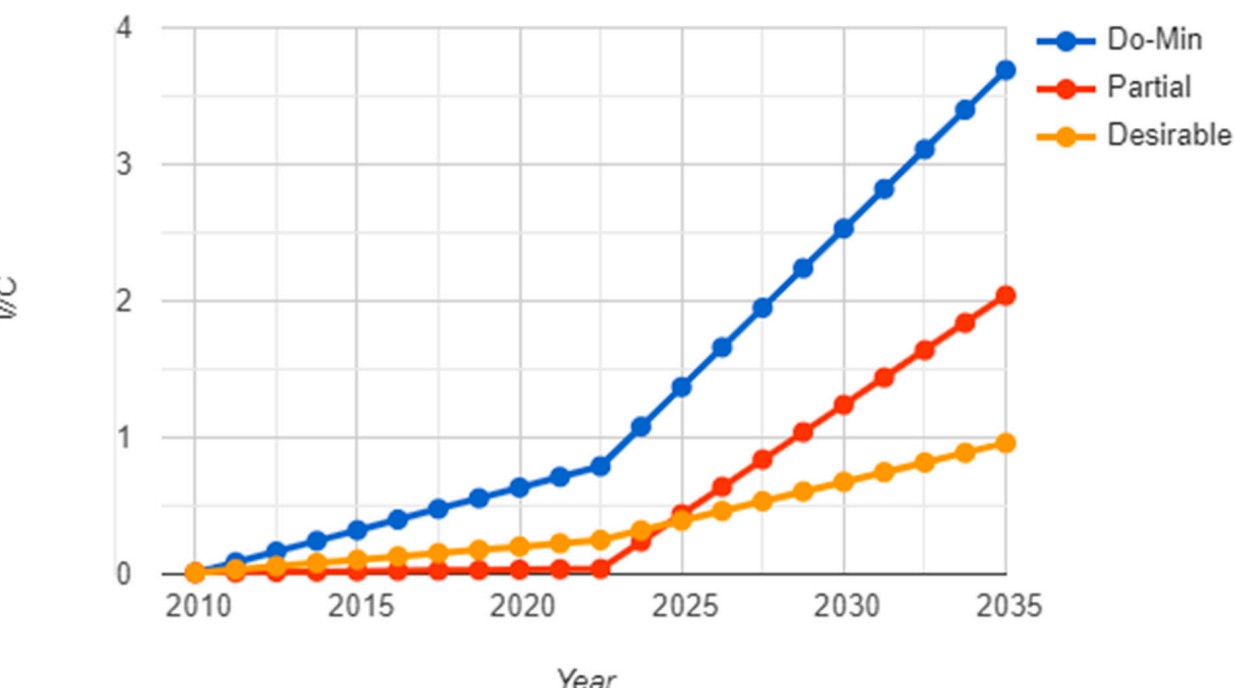

**Figure 20.** V/C ratio results for from population.

### 5.4. Demand and Supply

The demand and supply for transportation is projected by accounting for all the total supply of trips and the demand for trips for the desirable model. The different types of vehicles in motorized trips are considered, along with the public and private transportation during different months. Non-motorized vehicles are not considered. The total motorized trips are termed as demand, and private trips are termed as supply; the calculated D/S for the study area is given in Table 10.

**Table 10.** Study results for demand and supply.

| D/S | Do-Min | | | Partial | | | Desirable | | |
|---|---|---|---|---|---|---|---|---|---|
| Year | 2011 | 2021 | 2031 | 2011 | 2021 | 2031 | 2011 | 2021 | 2031 |
| D/S | 2.9 | 4.54 | 6.25 | 2.9 | 3.51 | 4.35 | 2.9 | 1.54 | 0.8 |

The projected values of D/S for the year 2031 are shown in Figure 21. The value starts at 2.9 in 2011; it is seen that the demand increases over the period for both the Do-Min and partial scenarios, while it reduces for the desirable scenario. The scenarios Do-Min and partial have similar ratios for demand and supply; for the desirable model, the data values start to decrease from nearly four to less than one. It is necessary to maintain this lower demand to supply ratio since it also affects the other variables.

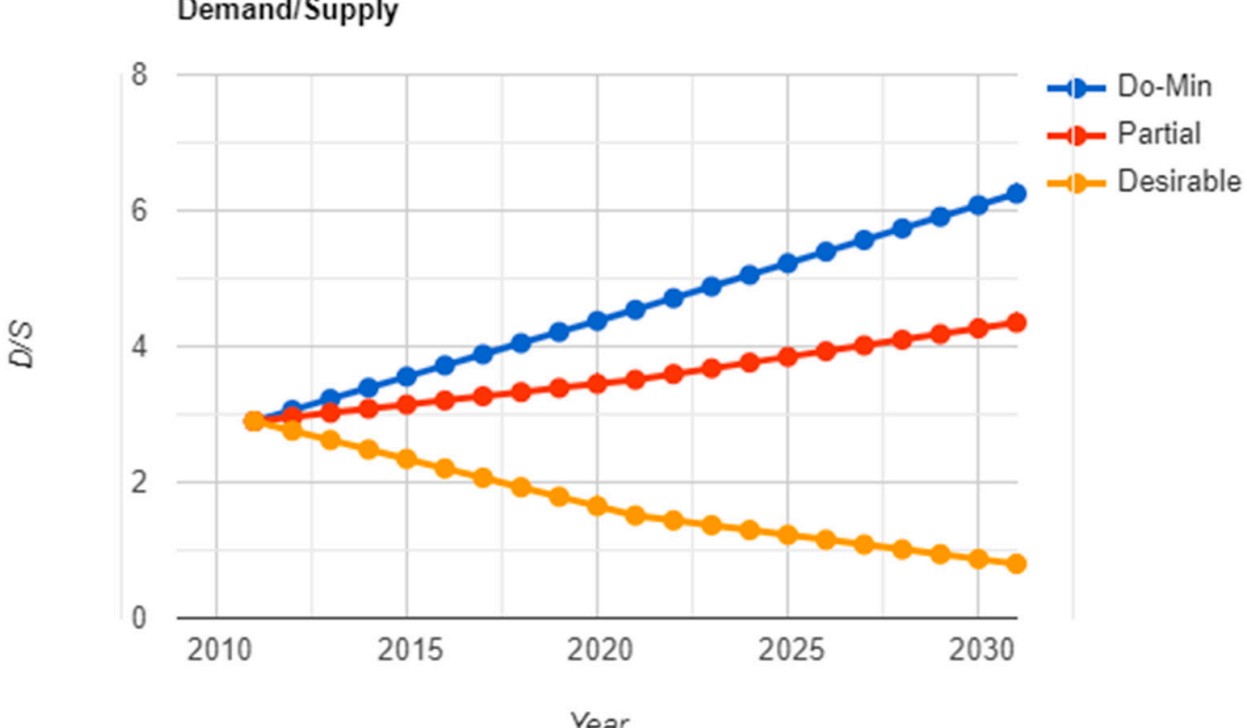

**Figure 21.** Results for demand/supply.

*5.5. Discussion*

The study created three scenarios, in which the Do-Min, partial and desirable proposals are implemented. For the Do-Min scenario, the existing trend is allowed to continue with no improvements. In the partial scenario, some new proposals implemented; however, most proposed models are ignored, hence, it is known as partial implementation of the model. The desirable scenario is the complete proposed model, in which all the proposals are implemented. In this model, the urban–rural population is much more balanced.

Practical Implications

The practical implications of these scenarios are identified and discussed here. The first scenario—Do-Min—allows the current trend to continue with no improvements. In this scenario, the population of the studied areas of Chennai grows at about five-fold. This is dangerous because unregulated population growth leads to sever social problems. The land usage is likewise spread unconditionally, with no trace of consistency or planning. Water bodies are turned into different sorts of structures, while fast urbanization results in high demand/supply in the transportation sector, high V/C of roads, unequal trip rates between rural and urban areas, and a large population. If the current trend continues, the V/Cs for 2011 and 2031 will be 1.28 and 3.69, respectively; the latter figure is 2.5 times greater than the base year. Similarly, the transport sector's D/S is 6.25 times that of the base year. According to this estimate, the number of cars is expected to increase at an alarming pace, causing further traffic congestion on the roadways. In addition, the demand for goods is outstripping supply. Such a scenario is unfavorable, and would have severe consequences in both the short- and long term. As a result, significant changes are necessary.

In the following scenario, some new ideas are implemented; however, not all suggested models are completed, resulting in incomplete model implementation. The FSI is used to manage land development; the permitted FSI in this case ranges from 1.5 to 2. The population is fairly managed, although there are significant disparities between rural and urban areas. The roads' V/C is 3.14, which is much greater than in the Do-Min scenario. The demand/supply ratio is also quite high, at 6.25 times that of the base year. Although the number of private automobiles has decreased, there has been no major change in the

number of vehicles. The number of automobiles on the road is likewise very disparate across urban and rural areas. This scenario shows that the vehicle density on the road has fallen marginally, as a result of the efforts taken to address the difficulties. Regardless of the adjustments, the D/S stays the same. As a result, the tiny number of adjustments made has minimal impact on the total outcome.

The final scenario is the desirable model, which takes into account all of the new recommendations. In this mode, the population is very evenly distributed between urban and rural areas. The rural and urban populations are growing at the same rate, indicating that both regions are developing in a balanced manner. The trip rate is advantageous due to both land use and population. Both the V/C and the D/S are less than one, indicating that the proposed model is very effective in controlling rapid urbanization, and that effective measures have been taken to improve the region's development. Based on the holding capacity of each region, the population density is kept at 500 people per hectare. The FSI has been fully developed and is set at three. In the transportation sector, an elevated corridor with increased capacity is proposed in the study area along the OMR; public transportation is doubled, and private transportation is cut in half to bring the LOS to the optimum level of V/C ratio, which is between 0.56 and 0.96. In the desirable scenario, the incorporated elevated corridor increases demand for public transportation through the introduction of metro and monorail services, and reducing private vehicles in a phase-wise manner; this achieves a dynamic balancing of land use sector holding capacity with carrying capacity. With a growth rate of 2%, the commercial, industrial, and institutional sectors are fully developed. In the transportation sector, a reasonable reduction in V/C ratio, from 1.28 and 3.69 in the Do-Min scenario to 0.56 and 0.96 in the desirable scenario, is achieved by incorporating growth restrictions that eliminate a certain percentage of cars and two-wheelers, and introducing an elevated corridor with a capacity of 3600 PCUs/hour, bypassing HCV and allowing only 30 percent of LCV. As a result, the desirable scenario's implementation outcomes are highly promising, and would significantly enhance the region's transportation and land quality.

## 6. Conclusions

In this paper, different parameters, such as population, land use, trip rate, V/C, and demand/supply, were simulated for three different years: 2011, 2021, and 2031. Three different scenarios were simulated: Do-Min, which is based on the continuing of existing trends; Partial, in which some proposals are implemented, and; desirable, in which all the proposals are implemented. The simulation was performed using the Stella simulation tool; from the results, it is identified that the desirable model would be highly effective in controlling the population and other parameters.

In the land use sector, a desirable density is achieved by restricting the population growth trend and augmenting the intensification of land use. The V/C ratio has an average difference of between 1 and 1.5. With respect to transportation, the growth of private transport is reduced by introducing better public transportation facilities. The novelty in this work is that the trips per day measure is calculated differently, which gives more accurate results than the existing methods. This work has taken the vertical development of the residential buildings into consideration, along with the TRpH. These variables are used as parameters for the calculation; these variables predict the traffic volume in the future and enable informed planning for future traffic demand. The outcomes of implementing the desirable scenario would greatly improve transportation and land quality in the region.

Future congestion can be reduced or avoided completely by improving the quality of life (QOL) of local people. Since most the new Chennai residents migrate from the rural areas, the desirable scenario suggests that decentralized development could improve all regions equally, thereby slowing down the migration to Chennai. The other reason for the movement to Chennai is the adequacy of investment in other cities. Hence, if the other cities became a focus for investment, migrants would increasingly migrate to their nearest cities, rather than to Chennai. From the research, it is seen that the proposed desirable

model would be highly effective in controlling the population and other parameters if implemented in real life. This research would also help researchers to analyze the traffic situation in the future, and make improvements to ease urbanization-related issues. If the desirable scenario is followed in real-time, it would resolve future problems that may arise from traffic congestion in Chennai.

**Author Contributions:** Conceptualization, Devi Priyadarisini K and G Umadevi; methodology, D.P.K.; software, D.P.K.; validation, D.P.K.; formal analysis, D.P.K.; investigation, D.P.K.; resources, D.P.K.; data curation, D.P.K.; writing—original draft preparation, D.P.K.; writing—review and editing, D.P.K.; visualization, D.P.K.; supervision, G.U.; project administration, G.U. All authors have read and agreed to the published version of the manuscript.

**Funding:** This research received no external funding.

**Informed Consent Statement:** Not applicable.

**Data Availability Statement:** The data that support the findings of this study are available from the corresponding author upon reasonable request.

**Conflicts of Interest:** The authors declare no conflict of interest.

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
