# Peer review of "A System Dynamics Model for Assessing Land-Use Transport Interaction Scenarios in Chennai, India"

_sustainability, doi:10.3390/su15076297_

Round 1

Reviewer 1 Report

Overall I like the work. The main, minor, comment is that there needs to be an overall statement of the lack of zoning, lack of permitting, and unchecked growth in the area.  This is broken up throughout the introduction, but in the first paragraph it basically states that big issue is inefficient use of space. It appears much more, requiring a master plan type approach that would be supported by your work.  This is just my initial thought, and you can move forward as you like, but the bigger problem does seem like there is not master plan, no zoning, no cohesiveness in the development that could be resolved with policy changes supported by your research.  

Furthermore, there needs to be some consistent theme the introduced and carried throughout the paper…items keep getting introduced, which is fine, but it is hard to follow the big complicated story. The abstract does a good job of this, it just needs to be repeated and expanded in the intro.  That stated, here are some specific comments:

Line 41:  Please elaborate on ‘unapproved’.  Is there no permitting system in place, no zoning, no master plan?  Can people just build? Pardon my ignorance on this region.  And given the unique configuration of the layout, could the authors provide an aerial impact of the region to give the reader a better idea of the location? I did a quick look at google maps/street view and can see the authors concerns.

Line 184….where are the policy recommendations? They should be in the conclusion, results, somewhere.

Line 273. Do_Min, Do Minimum, and Do-Minimum are found throughout the paper…be consistent with Do-Min and stick with it.

Figure1: Define D/S earlier in the body (not just the abstract).

Figure 2 How does the Do-Min, Partial, and Desirable fit in here (perhaps just adding them in the model)  Also, the three scenario idea should be introduced sooner.

Figure 5: What is ‘auto’ compared with the rest?

Fig 7/8:  Can’t tell difference between mixed res and res…perhaps a % for each legend item shown next to the color so Table 2 can be eliminated.  Also, what year?

Line 323.  Elaborate on the 1 sentence paragraph. What citations might be attributed to trip generation rate in this location??? Any manuals, previous work?

Table 7: What percent of the total area is changing? I.e. areas urban was 1338, down to 144, up to 4666?  This is confusing. To normalize this a bit for comparison the authors should consider showing percent of the total area…and report the total area in the caption.

Table 8:   There is a 300% growth in trip gens from population?  This needs further explanation when compared to 3.5% overall growth.

Conclusion: What is being proposed? What impact did the model demonstrate (high percent of trip growth, land use???)  Just sum of the work and the results. As a reviewer, I look at the abstract, Intro, and results and they need to have the same consistent story.

Author Response

Reviewer 1:

S.No

Queries

Reply/Justifications

1

Line 41: Please elaborate on ‘unapproved’. Is there no permitting

system in place, no zoning, no master plan? Can people just

build? Pardon my ignorance on this region. And given the unique

configuration of the layout, could the authors provide an aerial

impact of the region to give the reader a better idea of the location?

I did a quick look at google maps/street view and can see the

Zoning is ineffective in many residential areas of major cities in India, hence there are many unapproved sructures.

2

Line 184….where are the policy recommendations? They should

The policy recommendation will be done as a future scope for the next paper.

3

 Line 273. Do_Min, Do Minimum, and Do-Minimum are found

4

5

Figure 2 How does the Do-Min, Partial, and Desirable fit in here

(perhaps just adding them in the model) Also, the three scenario

6

Auto refers to auto-rickshaw and the same has been updated

7

Fig 7/8: Can’t tell difference between mixed res and res…perhaps

a % for each legend item shown next to the color so Table 2 can be

8

Line 323. Elaborate on the 1 sentence paragraph. What citations

might be attributed to trip generation rate in this location??? Any

This has been updated

9

Table 7: What percent of the total area is changing? I.e. areas

urban was 1338, down to 144, up to 4666? This is confusing. To

normalize this a bit for comparison the authors should consider

showing percent of the total area…and report the total area in the

We have made the required changes and given in percentage in table 7

10

Table 8: There is a 300% growth in trip gens from population?

This needs further explanation when compared to 3.5% overall

The overall growth is consistent, but it is not 3.5%.

11

Conclusion: What is being proposed? What impact did the model

demonstrate (high percent of trip growth, land use???) Just sum of

the work and the results. As a reviewer, I look at the abstract, Intro,

and results and they need to have the same consistent story.

The main purpose of the model is to bring a balance between holding and carrying capacity. the holding capacity denotes population and caring capacity denotes trips. And what would be the planning concepts to overcome the increased traffic in terms of trips for sustainable development.

the planning concepts (measures) are incorporated in the scenarios. And the growth are given in the model. All the growth rates taken in the model are taken to primary and secondary survey.

Reviewer 2 Report

The manuscript analysed the various factors of urban expansion and how these factors affect the transportation sector and traffic in Indian cities. The topic is interesting, but some issues should be addressed before moving to further step. 

1) In the literature review section, the current version only focus on reviewing the effects of unplanned urbanisation in Chennai and urban problems led by improper planning. I suggest the authors should add some paragraphs to review the impact of urbanisation on land use and transport, and then reveals the research gap. 

2) In the methodology section, the authors claimed the novelty in this work lies in the new attempt to determine the trips generated with respect to Chennai. With respect to Chennai is not the novelty, so what is the novelty of this paper? 

3) Regarding the Figure 2, it is unnecessary to show the flow of methodology, but it is important to make a chart to explain the methodology of this study.

4) In the discussion section, except for interpreting the three scenarios, the most important part is to answer what is the relationship between the growth rate of expansion and the impact on transportation. and how to make relevant planning recommendation and policy implication under the Indian context. 

5) The quality of figures and tables should be improved.  The current version have so many figures. 

Author Response

Reviewer 2:

S.No

Queries

Reply/Justifications

1

In the literature review section, the current version only focus on

reviewing the effects of unplanned urbanisation in Chennai and

urban problems led by improper planning. I suggest the authors

should add some paragraphs to review the impact of urbanisation

This has been updated in literature review

2

In the methodology section, the authors claimed the novelty in

this work lies in the new attempt to determine the trips generated

with respect to Chennai. With respect to Chennai is not the novelty,

It would be helpful for future policy makers

3

Regarding the Figure 2, it is unnecessary to show the flow of

methodology, but it is important to make a chart to explain the

4

In the discussion section, except for interpreting the three

scenarios, the most important part is to answer what is the

relationship between the growth rate of expansion and the impact

on transportation. and how to make relevant planning

  • It is concluded from the model results unless the investment level is increased and the required level of employment and infrastructure are created movement of the population towards metro sectors will not be to the desired level of achieving decongestion.
  • the study recommends that adopting the scenario III (IDEAL) of maximum investment and development as a policy measure would reduce the population, V/C, and pollution levels drastically when compared to all the other scenarios.
  • In the case of the existing area, the study reveals that not only by exercising control over the land use changes and regulating the land area development but combined with the augmentation of public transport alone, the land use intensity and transport capabilities get regulated. Simultaneously environmental pollution gets controlled and the level of service offered would get improved.
  • the simulation results of the present study thus establish the imperative need to identify the most probable dynamic changes expected during the foreseeable future in order to frame appropriate urban transport development policies.
  • the analytical model is framed to address the urban transport and land use system of a developing country and the simulation results have clearly established the functional validity model.
  • Application of this model to similar urban areas will prove to be valid with moderate modification on the structure of the model and this can be extended to micro-level planning also.
  • there is an imperative need to tie the kand use management system appropriately to the urban transport development programs.
  • land use planning can play a major role in Urban transport planning as a tool to ensure coherent development which would intergrate both land use activity and transport facilities.
  • It is vital to ensure that land use patterns are conducive to effective travel behavior.
  • the result suggests that there is an imperative need for a strong policy for promoting public transport in the study area.
  • it is importatant to link modal split and vechicle emission for enviromental sustainabilty.

5

The quality of figures and tables should be improved. The

Reviewer 3 Report

Dear editor,

   This article is interesting to tell something about urban India, while more refs should be added before its possible publication.  Please tell authors to make further improvements about their work.

Yours,

Miaoxi

Author Response

Reviewer 3:

S.No

Queries

Reply/Justifications

1

This article is interesting to tell something about urban India,

while more refs should be added before its possible publication.

Thank you. We have edited the document to enhance the content.